# TBX21 Methylation as a Potential Regulator of Immune Suppression in CMS1 Subtype Colorectal Cancer

**DOI:** 10.3390/cancers14194594

**Published:** 2022-09-22

**Authors:** Yuanyuan Shen, Yulia I. Nussbaum, Yariswamy Manjunath, Justin J. Hummel, Matthew A. Ciorba, Wesley C. Warren, Jussuf T. Kaifi, Christos Papageorgiou, Rene Cortese, Chi-Ren Shyu, Jonathan B. Mitchem

**Affiliations:** 1Institute for Data Science & Informatics, University of Missouri, Columbia, MO 65211, USA; 2Harry S. Truman Memorial Veterans’ Hospital, University of Missouri, Columbia, MO 65211, USA; 3School of Medicine, Washington University in St. Louis, St. Louis, MO 63130, USA; 4Department of Animal Sciences, University of Missouri, Columbia, MO 65211, USA; 5Department of Surgery, University of Missouri, Columbia, MO 65211, USA; 6School of Medicine, University of Missouri, Columbia, MO 65211, USA; 7Ellis Fischel Cancer Center, University of Missouri, Columbia, MO 65211, USA; 8College of Engineering, University of Missouri, Columbia, MO 65211, USA

**Keywords:** TCGA, scRNAseq, epigenetic, CD8+ T_EX_, TBX21, colorectal cancer, consensus molecular subtypes

## Abstract

**Simple Summary:**

Cytotoxic T lymphocytes (CTL) are critical for response to therapy and survival in CRC. Additionally, CTL are required for response to immune checkpoint inhibitor (ICI) therapy which does not work for most CRC patients. We utilized 7 omics datasets, integrating clinical and genomic data to determine how DNA methylation may impact survival and CTL function in CRC. Using comprehensive molecular subtype (CMS) 1 patients as reference, we found high TBX21 expression and low methylation had a significant survival advantage. To confirm the role of TBX21 in CTL function, we utilized scRNAseq data, demonstrating the association of TBX21 with markers of enhanced CTL function. Together, this study suggests that targeting epigenetic modification more specifically for therapy and patient stratification may provide improved outcomes in CRC.

**Abstract:**

Cytotoxic T lymphocyte (CTL) infiltration is associated with survival, recurrence, and therapeutic response in colorectal cancer (CRC). Immune checkpoint inhibitor (ICI) therapy, which requires CTLs for response, does not work for most CRC patients. Therefore, it is critical to improve our understanding of immune resistance in this disease. We utilized 2391 CRC patients and 7 omics datasets, integrating clinical and genomic data to determine how DNA methylation may impact survival and CTL function in CRC. Using comprehensive molecular subtype (CMS) 1 patients as reference, we found TBX21 to be the only gene with altered expression and methylation that was associated with CTL infiltration. We found that CMS1 patients with high TBX21 expression and low methylation had a significant survival advantage. To confirm the role of Tbx21 in CTL function, we utilized scRNAseq data, demonstrating the association of TBX21 with markers of enhanced CTL function. Further analysis using pathway enrichment found that the genes TBX21, MX1, and SP140 had altered expression and methylation, suggesting that the TP53/P53 pathway may modify TBX21 methylation to upregulate TBX21 expression. Together, this suggests that targeting epigenetic modification more specifically for therapy and patient stratification may provide improved outcomes in CRC.

## 1. Introduction

Colorectal cancer (CRC) is a heterogeneous disease characterized by distinct genome-wide changes with the third-highest incidence rate and the second-highest rate of cancer-related deaths worldwide [1]. To better design treatments for CRC, it is crucial to understand tumor heterogeneity and how this contributes to therapeutic resistance and disease progression. Genomic instability and epigenetic abnormalities with resultant dysregulation of gene expression are hallmarks of CRC. The high frequencies of DNA somatic copy number alterations (SCNAs) and APC tumor suppressor gene loss of function are closely linked with CIN-caused deletions, gains, translocations, and other chromosomal rearrangements, one of the primary pathways of CRC development [2]. Additionally, ~15% of CRC cases demonstrate alterations in DNA mismatch repair (MMR) proteins, which lead to hypermethylation and cancer development [3,4,5]. Importantly, a single factor does not lead to tumorigenesis, and this underlines the importance of understanding the molecular features of each individual tumor to improve therapy using a precision-based approach.

Immune-based therapies, such as immune checkpoint inhibition (ICI), have recently made significant advances in difficult-to-treat malignancies, such as non-small cell lung cancer, melanoma, and renal cell cancer [6,7]. However, these treatments are limited to patients with microsatellite instability-high (MSI-H) CRC, as they have not yet demonstrated efficacy in other patient groups [8]. Additionally, even in patients with an indication for use of ICI therapy, response is limited [9]. This is despite data demonstrating that the number of CTLs within the tumor microenvironment (TME) is a critical prognostic marker for CRC [10,11]. This again highlights the importance for improved methods of assessing the TME and understanding therapeutic resistance.

Dysregulated methylation impacts signaling pathways associated with apoptosis avoidance, metastasis, and therapeutic resistance and represents an important process in CRC [12]. Additionally, combined treatments with drugs targeting epigenetic modification exploit the dynamic nature of epigenetic changes to potentially modulate responses to immunotherapy [13]. However, current drugs targeting epigenetic modification are globally hypomethylating agents, such as Azacitidine and Decitabine, which are non-selective and may have unexpected effects [14]. Critically, a more specific understanding of epigenetic alterations is necessary to improve therapy and patient stratification.

Recently, large-scale public data repositories, such as The Cancer Genome Atlas (TCGA) and cBioPortal, as well as publicly available single-cell sequencing data, have allowed us to perform further in-depth study of cancer patients on a molecular and clinical basis using multi-omics [15,16,17]. To better classify CRC patients, the consensus molecular subtypes (CMSs) were developed by an expert panel using eighteen different CRC mRNAseq and microarray datasets, settling on four different subtypes based on molecular features [18]. Patients classified as CMS1 are characterized by microsatellite instability (MSI) and hypermutation and are considered the “immune” subtype. CMS2 is referred to as “canonical”, characterized by marked Wnt/β-catenin/TCF7L2 pathway activation and APC mutation. Characterized by metabolic deregulation, KRAS mutations, and mixed MSI patients, CMS3 is the least common. Tumors classified as CMS4 are “mesenchymal”, demonstrating prominent TGF-β activation, stromal infiltration, angiogenesis, and epithelial-mesenchymal transition (EMT) [18]. Recognizing the importance of molecular classification, there are ongoing pre-clinical trials utilizing this system for patient stratification and selection of immune-based treatments (NCT03436563, NCT04738214). To address limitations regarding the current knowledge in resistance to immune-based therapy and the role of epigenetic modification of gene expression, in this study we sought to combine multi-omics data with cutting-edge data analytics and comprehensive molecular classification using CMS (Figure 1). Utilizing publicly available data from four data repositories, including 15 datasets, 2391 CRC patients, and 7 omics datasets, we found a difference in survival based on differential expression and methylation of the critical T cell regulatory factor T-bet (TBX21). Additionally, we found that TBX21, MX1, and SP140 may play an important role impacting T cell function via TP53/P53 pathway modification of TBX21 methylation [19]. Together, this suggests that targeting epigenetic modification more specifically for therapy and patient stratification may provide improved outcomes in CRC.

## 2. Results

### 2.1. CMS1 Subtype Patients Demonstrate Features Consistent with Immune Activation

To address the question of the clinical impact of DNA methylation on gene expression and resistance to immune-based therapy, we chose to use TCGA, the largest publicly available multi-omics dataset with substantial clinical annotation. First, we downloaded and integrated the 461 colon cancer (COAD) and 171 rectal cancer (READ) patients from this dataset. We then classified patients in CMS subtypes to use the most comprehensive current molecular classification [18]. Using this system, there were 316 patients with DNA methylation data (450 k) and 509 patients with mRNAseq data labelled with CMS subtypes. The demographic, clinical, and pathologic characteristics of each patient group are summarized by CMS subtype in Table 1. As expected, the microsatellite status composition of patients was significantly different between CMS subtypes (*p* < 0.0001), with the CMS1 subtype containing more MSI-H patients and the CMS3 subtype containing more MSI-L/MSS patients. The median age of CMS1 patients’ diagnosis was 71, ranging from 58 to 85, significantly higher than patients in the other CMS subtypes (*p* = 0.0019). Additionally, in the CMS2 and CMS4 subtypes, more patients were diagnosed at stage III and IV than the other subtypes (*p* < 0.0001).

Next, we sought to combine CMS classification with stratification by cytotoxic lymphocyte (CTL) infiltration as a marker for anti-tumor immunity [20]. MCP-counter scores were derived for each patient, and patients were then stratified by MCP-counter score quartile and CMS classification (Figure 2) [21,22]. Patients in the CMS1 subtype had the highest median CTL abundance score, and most patients in this subgroup were in the highest quartile of CTL infiltration. Additionally, we also found that, when considering tumor mutational burden (TMB, Appendix A) and neoantigen-predicted peptides (Appendix A), CMS1 subtype patients were again much higher than the other subtypes. These data combined support the idea that CMS1 subtype patients represent an “inflamed” phenotype and are the most immune-active.

### 2.2. TBX21 Is the Only Gene Differentially Expressed, Differentially Methylated, and Highly Correlated with CTL Infiltration across All CMS Subtypes

CTL infiltration is critical for anti-tumor immunity and response to immune-based therapy. Therefore, we next sought to understand how gene expression and DNA methylation were associated with CTL infiltration and molecular alteration. Using CMS1 as the reference subtype, given its high level of CTL infiltration and predicted response to ICI, we performed differential gene expression and differential methylated region analysis (Figure 1). Expressions of 20,531 genes and 293,276 methylation loci were determined for each tumor sample from TCGA. There were 2977 differentially expressed genes and 11,541 differentially methylated regions (3500 differentially methylated genes) in the comparison of CMS1 and CMS2. In the comparison of CMS1 and CMS3, there were 2385 differentially expressed genes and 4231 differentially methylated regions (1583 differentially methylated genes). In the comparison of CMS1 and CMS4, there were 3246 differentially expressed genes and 9393 differentially methylated regions (2359 differentially methylated genes). To identify “crosstalk genes”, genes both differentially expressed and differentially methylated, we then performed a correlative analysis plotting differentially expressed (|logFC| > 1 and *p*-value < 0.05, adjusted for FDR) against differentially methylated genes (|Δβ| > 0.25 and *p*-value < 0.05, adjusted for FDR, Figure 3a–c). Identified crosstalk genes were predominantly enriched in quadrants “higher methylation and higher expression” and “higher methylation and lower expression” (Figure 3a–c), demonstrating increased methylation (Δβ > 0.25) associated with either increased or decreased differential gene expression. To validate these findings, we analyzed other non-TCGA publicly available gene expression and DNA methylation datasets (Appendix A) [23]. Analysis of these datasets confirmed that TBX21 was consistently differentially methylated when comparing CMS1 vs. other CMS subtypes. Specifically, we found that one loci in the coding region of TBX21 (SiteID: cg26281453, Loc: 45810610) was differentially methylated in all datasets (Appendix A).

To explore the relationship of crosstalk genes with CTL infiltration, we correlated crosstalk gene expression with MCP-counter CTL abundance scores. Interestingly, TBX21 was the only gene differentially expressed, differentially methylated, and highly correlated with CTL from each CMS1 comparison (r > |0.7|, Figure 4a–c). Interestingly, we observed that TBX21 was more highly expressed in CMS1 patients but was also more highly methylated (Figure 3a–c). However, the typical expectation is that methylation is an epigenetic modification that leads to decreased gene expression [5]. Moreover, previous studies have suggested that TBX21 is an important transcriptional regulator of tumor-reactive CD8T-cells, which are critical for response and survival [24]. Given these results, we hypothesized that the importance of alterations in the expression of TBX21 via methylation may be highest within CMS1 patients.

### 2.3. CMS1 Patients with High TBX21 Expression and Low TBX21 Methylation Have the Best Survival

To better understand the relationship between TBX21 expression and methylation and CMS1 patient outcomes, we next performed a survival analysis. When using TBX21 expression or methylation values as independent variables, there was no difference in survival. Therefore, we further classified CMS1 patients using the median value of TBX21 expression and mean methylation β value to separate these patients into four groups (Figure 5a). Patients were grouped as: high TBX21 expression with low methylation (high-low); high TBX21 expression with high methylation (high-high); low TBX21 expression with high methylation (low-high); and low TBX21 expression with low methylation (low-low). We then repeated the survival analysis with CMS1 patients stratified by these four subgroups (Figure 5b). Patients classified as high-low (high expression, low methylation) demonstrated the best survival, followed by low-high patients. Interestingly, the group of patients classified as high-high appeared to represent a potential intermediate subgroup (Figure 5a) with worse survival (Figure 5b) than both high-low and low-high subgroups. These data suggest that the interaction between TBX21 expression and methylation plays an important role in patient prognosis.

### 2.4. There Were No Significant Clinical Differences between High-Low and Low-High CMS1 Patient Subgroups

This result inspired us to investigate the potential mechanisms of the observed difference in survival between CMS1 patient subgroups. Therefore, we retrieved the patient clinical attributes from cBioPortal, including 41 separate attributes. We found no clinical attributes with significant differences when comparing high-low and low-high patients. However, when comparing the high-high group to other groups, we found multiple significant factors. There were eight clinical attributes with significant differences between high-low and high-high patient groups (Appendix A). Most importantly, we noted that high-high patients had a significantly higher number of positive lymph nodes than high-low patients (*p*-value: 0.0442, Figure 6a), a known marker of poorer survival. Additionally, high-high patients had significantly higher rates of lymphovascular invasion (LVI) than low-high patients (*p*-value: 0.0192, Figure 6b), another important clinical risk factor for survival. These results suggest that the survival difference demonstrated by the high-high group may be driven by clinical factors; however, the survival difference between the high-low and low-high patient groups may be more likely driven by molecular factors.

### 2.5. Patients with high TBX21 Expression and Low Methylation Are the Most Highly Immune-Infiltrated

To further evaluate the impact of TBX21 on survival and anti-tumor immunity, we looked at other indicators of immune resistance. Tumor mutational burden (TMB) has been shown to predict survival and response to immune checkpoint, so we first derived these scores for each patient [25]. When comparing the subgroups of CMS1 patients, however, we observed no significant differences in TMB (Appendix A). Next, we derived neopeptides, a potentially better marker of immune reactive antigen in the tumor microenvironment, and, again, we did not observe any differences between subgroups of CMS1 patients (Appendix A). Given the lack of overt differences in these measures, we sought to further investigate other aspects of anti-tumor immunity.

To obtain a more in-depth look at immune alterations in CMS1 patient subgroups, we next looked more specifically at different cell populations in the TME using cell marker score analysis [26]. Signature genes for calculating the cell marker score of each cell population were obtained from the TCIA dataset (Appendix A) [26]. Using this analysis, we found that high-low patients had significantly higher infiltration of CD8+ T cell subtypes, including activated CD8+ T cell (*p* value = 0.0008), central memory CD8+ T cell (*p* value = 0.0027), and effector memory CD8+ T cell (*p* value = 0.0023) (Figure 7a–c). We also found increased infiltration of T helper cells (Th1, *p* value = 0.0007) and activated dendritic cells (DC, *p* value = 0.0007) (Figure 7d,e). However, in addition to the significantly higher infiltration of cell subtypes consistent with anti-tumor immune profiles, we also saw increased infiltration of immune-suppressive cell subtypes, such as Treg (*p* value = 0.0040) and myeloid-derived suppressor cells (*p* value = 0.0027) (Figure 7f,g). TH17 and monocyte subsets were not significantly different. Given the evidence of increased immune cell infiltration, both pro-and anti-inflammatory, we sought to further understand the molecular differences that may suggest a mechanism for the observed difference in survival associated with TBX21 in these patients.

### 2.6. Epigenetic Modification of Genes in the Regulation of TP53 Activity including TBX21 Are Enriched in High Expression and Low Methylation Patients

To look at the question of molecular differences impacting survival in CMS1 patient subgroups, we completed DEG analysis, focusing on the differences between high-low and low-high patients (Table 2). Notably, we found that there were many more genes differentially expressed when comparing high-low and low-high patients than when comparing the other subgroups (1482 DEGs, Appendix A), further supporting an important molecular difference in these patient groups. Next, DEGs obtained from comparing high-low and low-high subgroups were analyzed using Reactome gene enrichment analysis. We found 741 DEGs that were enriched in 41 pathways (Appendix A). Eighteen CD8+ T_EX_ marker genes were enriched in thirteen pathways, all of which were upregulated in high-low patients (Figure 8). After processing DMG analysis between low-high and high-low patients, five crosstalk genes associated with the CD8+ T_EX_ signature were identified (Appendix A). We next applied Reactome analysis specifically for crosstalk genes. These five crosstalk genes participated in 303 pathways (Appendix A). Notably, we observed that genes MX1, SP140, and TBX21 were frequently enriched in the “Regulation of TP53 Activity” and its cascade pathways. Moreover, we identified the function of SP140 from the EpiFactors database as a Zinc finger structure that mainly targets Histone modification read and transcription factor (TF) regions to participate in epigenetic modification [27]. Together, these data suggest a critical role for epigenetic modification of TP53 pathway genes impacting TBX21 and patient outcome in these subgroups.

### 2.7. TBX21 Is a Key Modulator in CD8+ T Exhausted Cells

To validate our hypothesis regarding the importance of TBX21 in T_EX_ in CRC, we explored publicly available scRNAseq data from 23 CRC patients with 65,362 matched normal and tumor single cells. Data were first normalized and then, utilizing cell subtypes identified by the original study, we retrieved 47,285 tumor cells labeled as TP [7]. Blueprint/ENCODE references from the SingleR subtype identifier were then used to re-annotate tumor cells [28]. We identified 36 pure stroma and immune cell types, including 2268 central memory CD8-positive alpha-beta T cells (CD8+ T_CM_); 4214 effector memory CD8-positive alpha-beta T cells (CD8+ T_EM_); and 84 CD8-positive alpha-beta T cells (CD8+ T cells). To focus on exhausted CD8+ T cells, we retrieved all CD8+ T cell subsets (CD8+ T cells) and performed an independent cluster analysis. In this analysis, we found eight distinct CD8+ T cell subclusters, each exhibiting a distribution of clusters. To annotate these clusters, we then used CellMarker, a marker-based annotation database, and identified clusters 1 and 7 as CD8+ T exhausted (CD8+ T_EX_) cells (Appendix A) [29]. Compared with the pre-identified annotation from the Blueprint/ENCODE references, CD8+ T_EX_ predominantly overlapped with CD8+ T_EM_ cells and a small proportion of CD8+ T_CM_ cells (Figure 9a,b). Additionally, we selectively looked at the expression of transcription factors, checkpoint receptors, and effector molecules, noting that, in subclusters of CD8+ T_EX_, cluster 7 was enriched with cells that had a significantly higher average expression of MKI67 and PDCD1 and lower expression of TBX21 than cluster 1 (Figure 10) [29]. Cluster 1 demonstrated expressions of cells that were consistent with the identified low-proliferative T_EX_ cluster in the previous research and consistent with the idea that TBX21 expression is associated with more highly functional cells [29]. In contrast, cluster 3 contained cells with the highest TBX21 expression and low levels of expression of checkpoint receptors, such as PDCD1, LAG3, and TIGIT, when compared with CD8+ T_EX_ (clusters 1 and 7) (Figure 10 and Figure 11). We then created feature plots to show the distribution of TBX21, PDCD1, and EOMES. Specifically, PDCD1 showed the highest density in CD8+ T_EX_ (clusters 1 and 7) and was associated with CD8+ T_EM_ (Figure 9a and Figure 12), and there was minimal overlap in cells expressing TBX21 and PDCD1. To further confirm our findings in the bulk mRNAseq and DNA methylation data, we then looked to see if the expressions of MX1 and SP140 were associated with TBX21 expression and found that the expression of these genes was associated with TBX21 expression, supporting our findings. Together, these data suggest strongly that MX1 and SP140 utilize epigenetic modification and cooperate with TBX21 through the TP53 cascade pathways to decrease expression of T_EX_ cell markers and improve function in CD8+ T cells.

## 3. Discussion

DNA methylation plays an important role in the development of CRC; however, its potential role in immune dysfunction is less well characterized [5]. This may be most important in the subgroup of patients that demonstrate hypermethylation, MSI-H patients. These are the patients most likely to respond to immune checkpoint inhibition and are the only CRC patients with a current FDA-approved indication for ICI therapy [30]. However, MSI status has been shown to be an incomplete marker, leading to the establishment of CMS subtypes to better characterize this heterogeneous disease on a molecular level [31]. To better understand the impact of DNA methylation on immune dysfunction and potential resistance to immune-based therapy in CRC patients, we integrated data from TCGA with cutting edge bioinformatic techniques demonstrating that, in CMS1 patients, the “inflamed” subtype of CRC, TBX21 methylation, and expression stratified patients into groups with significantly different survival. A known critical transcriptional factor in T cell function, this suggested an important role for methylation of TBX21 in CRC [24]. Using further in-depth analysis validated in publicly available scRNAseq data, we found evidence that TP53 pathway genes MX1 and SP140 may participate in the epigenetic modification of TBX21 and impact patient survival.

Unlike the canonical pattern of epigenetic modification, TBX21 demonstrated both increased methylation and expression (high-high) in CMS1 patients. This may primarily be related to the majority of CMS1 patients being MSI-H. However, this implied that methylation of TBX21 was most important in CMS1 patients. Previous research suggests that genes in the high-high quadrant have a more complex and dynamic manner of regulation of gene expression by DNA methylation, especially during carcinogenesis and metastasis [32]. Our results identified that hypermethylated loci in TBX21 are mainly enriched in CTCF, a promoter, and some coding areas in CMS1 patients. This suggests that hypermethylation of selective CpG islands in the TBX21 promoter region contributed to the observations in this study.

There is significant research linking TBX21 with T cell exhaustion; therefore, it fits that this gene would impact survival in the subset of CRC patients characterized by an active anti-tumor immune response. Early studies first demonstrated that a gradient of TBX21 expression led to direction of CD8 T cells towards short-lived, highly active effectors versus long-term “slow burn” effectors in response to viral illness [33]. Further work has then developed the story into showing clearly important roles for TBX21 in the regulation of interferon-γ production by regulating the accessibility of the IFNG gene through chromatin remodeling in the context of infection [34]. In another murine infection model, others further demonstrated that TBX21 appeared to be a critical regulator of PD-1 expression and was susceptible to epigenetic disruption impacting CD8 T cell exhaustion [35]. Most recently, Beltra et al. published an elegant description of the impact of transcriptional alteration of TBX21 and TOX on CD8 T cell exhaustion where they also demonstrated that the CTL exhaustion depicted in chronic viral illness correlates with similar makers in CTL from a small group of patients with melanoma [36]. In this study, we further build on the data exhibited by Beltra et al. by demonstrating that TBX21 expression and epigenetic modification have an impact on patient outcome in CRC, which has not previously been shown. Additionally, we connect the work of Barili et al. by showing that epigenetic alteration of TP53 pathway genes is dysregulated in conjunction with TBX21, suggesting novel therapeutic combinations that may work to improve outcomes in “inflamed” CRC [37]. Despite the substantial improvement in outcome that has been seen with the application of ICI therapy in these patients, the most recent data suggest that only 40% of patients demonstrate therapeutic response [9]. Further work is critical in this area to fully elucidate the mechanistic relationship of TBX21 methylation and survival in CRC patients.

Although we attempted to control for weaknesses, our study suffers from some significant limitations. First, as with any retrospective analysis of clinical data, this study was subject to bias based on clinical factors. Additionally, while TCGA is the most robust publicly available dataset including multi-omics and clinical data, our analysis suffers from limitations due to patient numbers in the specific subgroups [38]. Furthermore, due to the nature of the data, we were unable to review patient records for accuracy and further explore the potential impact of confounding variables on outcomes of interest. We attempted to compensate for this through comparative analysis of clinical factors associated with survival, noting no differences between the high-low and low-high CMS1 patients in univariate analysis. To validate our findings, we used other publicly available methylation datasets; however, there were no datasets in which to confirm all our findings, particularly in the context of CMS subtyping. To test our hypothesis regarding the importance of TBX21 in CTL function, we utilized a large publicly available scRNAseq dataset, but we were unable to directly explore TBX21 expression and methylation at the single cell level in CRC using existing available data. Additionally, these data were not associated with patient outcome and, therefore, could not directly address the impact of TBX21 expression, T cell function, and patient outcome. This represents an exciting area of future exploration utilizing advanced single cell techniques [39].

In this study, we integrate comprehensive scRNAseq, mRNAseq, methylation, TMB, neoantigen, clinical and MCP-counter scores from multiple datasets to explore the role of TBX21 in CD8+ T cell exhaustion and patient outcome in CRC. We demonstrate that, in CMS1 subtype CRC patients, those with high TBX21 expression and low methylation have improved survival, suggesting an important role for epigenetic regulation of TBX21 in the outcome of these patients. Moreover, epigenetic modification of TBX21, along with MX1 and SP140, may provide this impact via the TP53/P53 pathway. Therefore, DNA methyltransferase inhibitors combined with immune checkpoint blockade may further support CTL function in CMS1 patients via protection of TBX21 expression. Future work focusing on enrolling sufficient patients with strict quality control and application of cutting edge biomedical informatics techniques at the single cell level is required for further understanding of DNA methylation and gene expression in CD8+ T cell function and patient outcome [40].

## 4. Materials and Methods

In this study, we utilized 15 datasets, 2391 CRC patients, and 7 omics datasets and integrated comprehensive clinical and genomic data (epigenetic, scRNAseq, mRNAseq data, mutations), well-accepted immune infiltration indicators (TMB, neoantigen), and cell marker and MCP-counter scores from multiple datasets to reveal that TBX21 is a critical regulator in CD8+ T cell exhaustion (Figure 1).

### 4.1. Data Preparation

We downloaded 10× scRNAseq raw_UMI_count_matrix and cell_annotation data from Gene Expression Omnibus (GEO) with the accession number GSE132465 [41]. Bulk mRNAseq, DNA methylation (450 k), and mutation data of CRC were collected from TCGA. Moreover, we used publicly available data deposited in the ArrayExpress database at EMBL-EBI under accession numbers E-MTAB-7036 (methylation) and E-MTAB-8148 (microarray), and additional microarray data from Colorectal Cancer Subtyping Consortium (GSE13067, GSE13294, GSE17536, GSE20916, GSE33113, GSE37892, GSE39582, KFSYSCC) were stored under the synapse repository (https://www.synapse.org/#!Synapse:syn2623706/files/, accessed on 28 March 2020) [17]. For each CRC patient, corresponded mutations (per Mb), neoantigens with HLA alleles, and mutated expressed peptides were retrieved from The Cancer Immunome Database (http://tcia.at, accessed on 28 June 2020) [26].

### 4.2. Demographics and Clinicopathologic Analysis

We integrated pertinent clinical data (age, gender, microsatellite status, pathologic stage, days to last follow up, and vital status), mRNAseq, and DNA methylation (450 k) by participant ID. Each CRC patient had pre-identified CMS subtypes from the original study [18]. Due to missing information, including indeterminate microsatellite status and unclear CMSs, eventually, we identified 316 patients with DNA methylation data (450 k) and 509 patients with mRNAseq data with distinct CMS labels. We then grouped DNA methylation data and mRNAseq data by CMSs. Statistical analyses based on grouped patients were performed in Prism 7 (GraphPad Software). Patients’ basic clinical features were summarized by descriptive statistics, including means and standard deviation, and an unpaired *t*-test and Mann–Whitney test were used for normally distributed continuous data. Categorical variables were compared using Fisher’s exact and chi-square tests. A *p*-value < 0.05 was considered statistically significant.

### 4.3. mRNAseq, Microarray Differential Gene Expression, and DNA Methylation Differential Region Analysis

mRNAseq differential gene expression analysis was performed with the edgeR package using the raw data downloaded from the TCGA dataset Illumina-HiSeq and TCGA_Illumina-GA platforms [42]. Differentially expressed genes were defined as genes with an |logFC| > 1 and *p*-value < 0.05 for comparisons of CMS1 vs. CMS2, CMS1 vs. CMS3, CMS1 vs. CMS4, and CMS1 patients with high-low vs. low-high. Genes with Benjamini–Hochberg-adjusted false discovery rate (FDR) < 0.05 were considered to be significantly differentially expressed for further steps. For each cohort, we identified 20,531 total genes by mRNAseq raw counts.

The microarray data obtained from EMBL-EBI and Colorectal Cancer Subtyping Consortium had CMSs defined from the original study. We grouped the patients by CMSs for DEG analysis. Probe identification numbers were converted into gene symbols. The Affy package was used to normalize gene expression values [43]. The Limma R package was applied to identify the DEGs between CMS1 and other CMSs [44]. *p* values < 0.05 and adjusted *p* values < 0.05 were used as the DEG cutoffs.

COHCAP R package was used to identify differentially methylated regions (DMRs) [45]. Δβ and *p*-value between every two groups were calculated based on β-values of CpG sites using COHCAP. DMRs were defined as methylated regions with |Δβ| > 0.25 and *p*-value < 0.05 (hypermethylated DMRs: Δβ > 0.25 and *p*-value < 0.05; hypomethylated DMRs: Δβ < 0.25 and *p*-value < 0.05). Gene symbols were annotated based on DMRs using the annotation files of methylation profiling. DMRs corresponding to no gene symbol and multiple gene symbols were not obtained for further analysis. Genes with DMRs were defined as differentially methylated genes (DMGs).

### 4.4. MCP-Counter Score Processing

Then, for every patient, we implemented the R package Microenvironment Cell Populations (MCP)-Counter in the mRNAseq using Expectation-Maximization (RSEM) normalized mRNAseq data to create cell-type abundance scores [21]. Ten cell populations were simultaneously quantified in the tumor microenvironment, including eight immune cell populations (T cells, CD8 T cells, cytotoxic lymphocytes, NK cells, B lineage, monocytic lineage, myeloid dendritic cells, neutrophils), endothelial cells, and fibroblasts. Finally, we used a heatmap from the ggplot2 R package to visualize the percentile of MCP-Counter scores for each CMS and cell population [46].

### 4.5. Pearson’s r Correlation Coefficient

We identified the DEGs from the mRNAseq DEG analysis and methylation DMGs as crosstalk genes from each comparison. Pearson’s r correlation coefficient was used to detect correlations between crosstalk DEGs’ mRNAseq expression and cytotoxic lymphocytes scores and |r| ≥ 0.7 was used to determine the highly correlated genes.

### 4.6. Survival Analysis

We separated patients into four subgroups based on the median of TBX21 expression and the mean of the TBX21 methylation β value. The Kaplan–Meier survival curves generated were assessed with the Cox regression model for each immune functional differentially expressed gene using Prism 7. The survival curves between each subgroup of patients were compared by log-rank test. Overall survival was used as the endpoint for each patient, either the days from diagnosis to death or the last follow-up.

### 4.7. Pathway Enrichment Analysis

Pathway enrichment analysis was performed to evaluate the pathways associated with differentially expressed genes of high-low vs. low-high in CMS1 patients. We highlighted the DEGs on the bar chart. These DEGs were also markers of CD8+ T_EX._ A barplot was used to illustrate the comparison of enriched reactome pathways among differentially expressed genes [47]. These results were analyzed with the clusterProfiler, DOSE, and ReactomePA R packages [48,49,50].

### 4.8. Tumor Mutation Burden, Neoantigen Analysis, and Cell Marker Score Calculation

TMB is a measure of the total number of mutations per megabyte of tumor tissue. The mutation density of tumor genes is also defined as the average number of mutations in the tumor genome, including the total number of gene coding errors, base substitution insertions, or deletions. The value of 38 Mb is routinely taken based on the length of the human exon, so the TMB estimate for each sample is equal to the total mutation frequency/38. TMB per megabase is calculated by dividing the total number of mutations by the size of the coding region of the target.

We retrieved neoantigen data from TCIA [26]. For each patient, we counted the number of genes, neopeptides generated by each gene, and the uptake HLA alleles. We applied ordinary one-way ANOVA for the statistical analysis.

We utilized mRNAseq expression of identified signature genes for each immune cell type. The geometric mean formula was applied to calculate the cell marker score, which is the value of each signature gene’s normalized mRNAseq expression for each cell population. Machine learning-based methods identified these cell marker genes based on all TCGA patients and other large datasets from the TCIA database [26].

### 4.9. scRNAseq Data Analysis

All R packages were used to process the 10× scRNAseq data analyses under R version 4.0.3. SingleCellExperiment and scatter R packages were used to integrate cell_annotation and raw_UMI_count_matrix data, as the SingleCellExperiment object was collected from the GEO [51]. After removing genes not expressed in any cell, we normalized the SingleCellExperiment object by log2-transformation. Principal component analysis (PCA) was performed based on the 27,953 genes to analyze the normalized object. Based on the available annotations from the original study, we subset the “Tumor” from the “Class” annotation label, including 47,285 CRC cells. R package with Blueprint/EncodeData reference was applied to annotate immune cell types [52]. Next, we collected cell subtypes annotated as “CD8+ T-cells”,”CD8+ Tcm”, and “CD8+ Tem”, as we as overall CD8+ T-cells. Seurat R package was used to convert the SingleCellExperiment object to a Seurat object, and the “FindVariableFeatures” function was used to select the top 2000 highly variable genes [53]. The “FindClusters” function with a resolution of 0.5 was applied to the Seurat object following CD8+ T-cell clustering. Uniform manifold approximation and projection (UMAP) were applied to explore the subclusters. After identifying the CD8+ T-cells subclusters, the “FindAllMarkers” function was used to define highly differentially expressed genes between clusters. Moreover, the “Idents” and “Simplot” functions were utilized to visualize the overlap between annotated CD8+ T-cell subtypes with CD8+ T-cell subclusters. The CellMarker dataset was applied to recognize the CD8+ T-cell subclusters by highly differentially expressed genes [29].

## Figures and Tables

**Figure 1 cancers-14-04594-f001:**
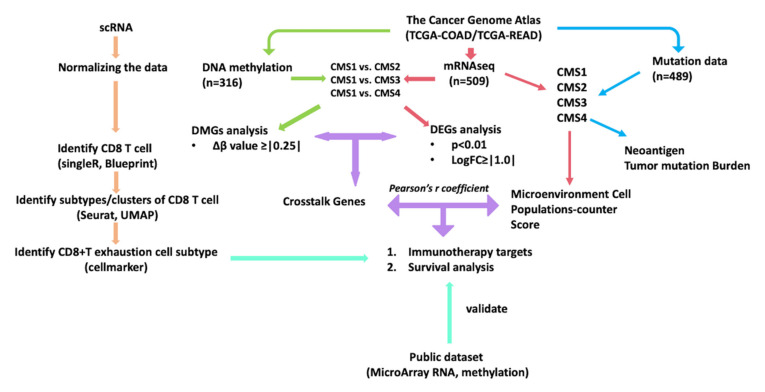
An outline of the methods and organization of this study. This flowchart describes the multi-omics we included and the process of how we breakdown patients by CMS classification. We then compared CMS1 and other CMS subtypes to determine DEGs and DMGs. Then, the MCPcounter score was calculated and survival analysis was undertaken using the identified CD8+ T exhaustion cell maker and immuno-crosstalk gene. Pathway associations were determined for DEGs using the Reactome online browser. Tumor mutation burden and neoantigen were used to evaluate the CMS subtypes. Eleven publicly available datasets were used to validate our partial results. scRNA, single-cell RNA-Seq; DMG, differentially methylated gene; DEG, differentially expressed gene.

**Figure 2 cancers-14-04594-f002:**
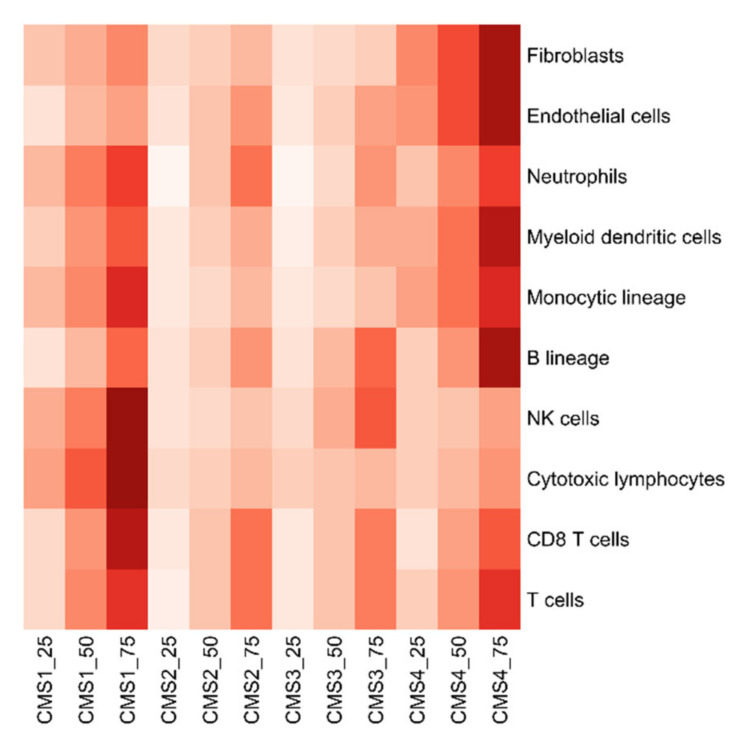
Ten tumor microenvironment cell populations for MCP-counter scores. CMS1 patients have significantly higher cytotoxic lymphocyte scores than the others. Each set of three columns represents one CMS subtype, each row represents the cell population.

**Figure 3 cancers-14-04594-f003:**
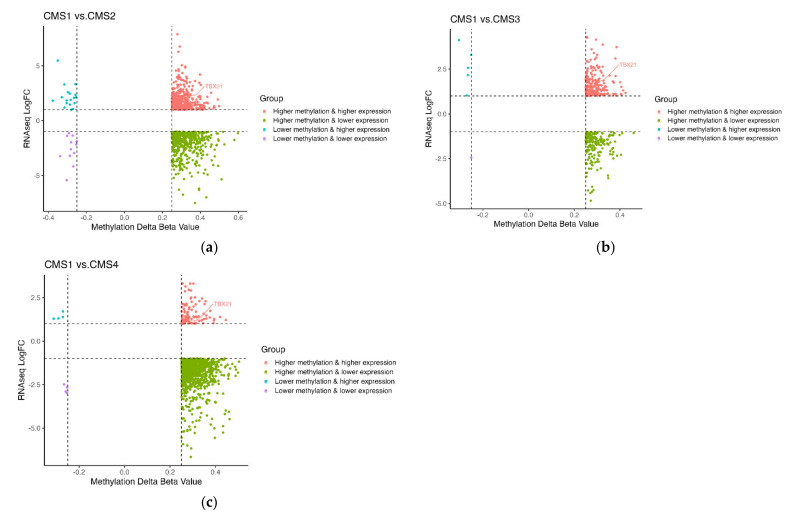
The distribution of crosstalk genes. Among the identified crosstalk genes for each comparison, most crosstalk genes were enriched in quadrants 2 and 3. TBX21 was the only gene differentially expressed and differentially methylated across three comparisons. The *x*-axis is the delta beta value of methylation, the *y*-axis is the RNAseq LogFC. (**a**) Distribution of crosstalk genes from CMS2 vs. CMS1 DEG and DMG analysis. (**b**) Distribution of crosstalk genes from CMS3 vs. CMS1 DEG and DMG analysis. (**c**) Distribution of crosstalk genes from CMS4 vs. CMS1 DEG and DMG analysis.

**Figure 4 cancers-14-04594-f004:**
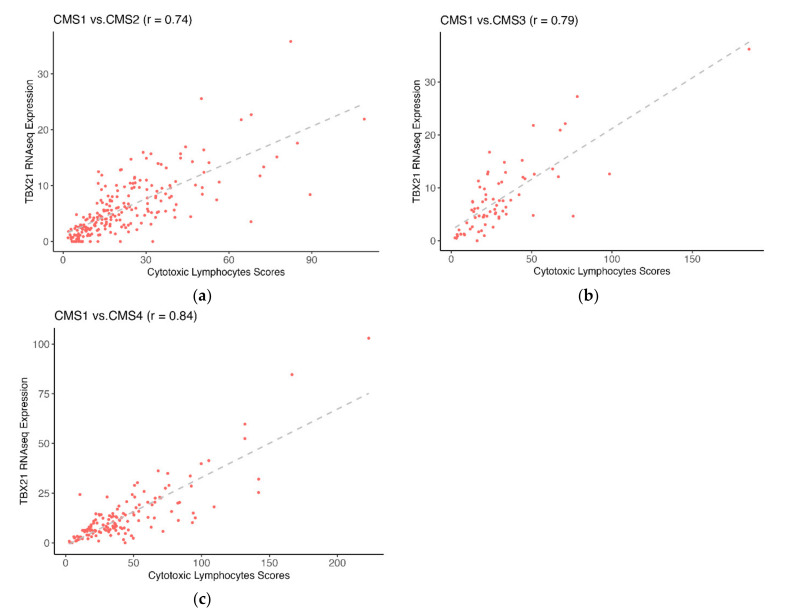
TBX21 was highly correlated with CTL. The x-axis is the CTL score, the y-axis is the normalized TBX21 mRNAseq expression. (**a**) Normalized TBX21 expression was highly correlated with CTL in CMS2. (**b**) Normalized TBX21 expression was highly correlated with CTL in CMS3. (**c**) Normalized TBX21 expression was highly correlated with CTL in CMS4.

**Figure 5 cancers-14-04594-f005:**
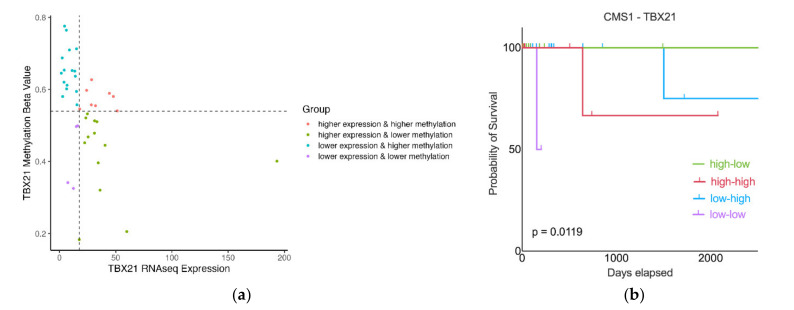
(**a**) Subtype of CMS1 patients based on the expression and methylation of TBX21. There were 24 patients with higher TBX21 methylation and 17 patients with lower methylation. The *x*-axis is TBX21 mRNAseq expression, the *y*-axis is the TBX21 methylation beta value. High-low: high TBX21 mRNA expression, low TBX21 methylation (number of patients (n) = 13); high-high: high TBX21 mRNA expression, high TBX21 methylation (n = 8); low-high: low TBX21 mRNA expression, high TBX21 methylation (n = 16); low-low: low TBX21 mRNA expression, low TBX21 methylation (n = 4). (**b**) Patients with higher expression and low methylation of TBX21 had the best survival, and the patients with lower expression and higher methylation had worse survival than high-low group and there were also the largest number of patients with this status in this group. The *x*-axis is patient days elapsed, the *y*-axis is percent of patients still alive.

**Figure 6 cancers-14-04594-f006:**
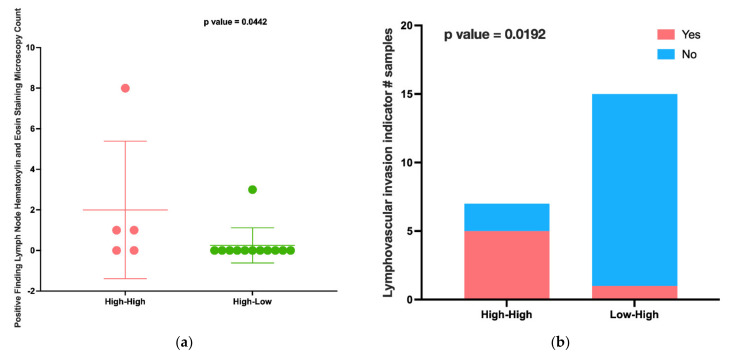
(**a**) High-high patients had more positive lymph nodes than high-low patients. (**b**) high-high patients had more lymphovascular invasions than low-high patients.

**Figure 7 cancers-14-04594-f007:**
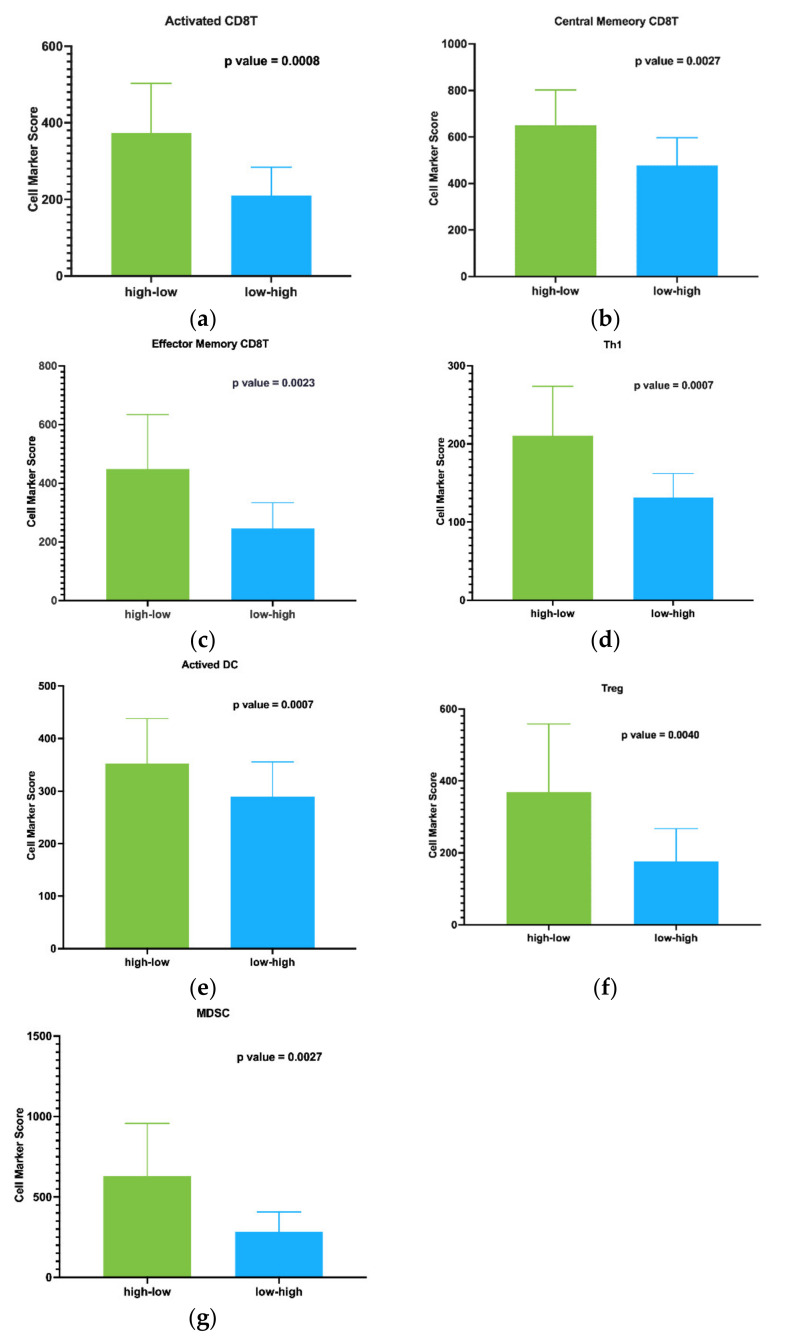
Cell marker score analysis for immune cell populations. High-low patients had significantly higher infiltration of CD8+ T cell subtypes, including (**a**) activated CD8+ T cell, (**b**) central memory CD8+ T cell, and (**c**) effector memory CD8+ T cell. We also found increased infiltration of (**d**) T helper cells (Th1) and (**e**) activated dendritic cells (DC). We also saw increased infiltration of cell subtypes suggested to be immune-suppressive: (**f**) Treg and (**g**) myeloid-derived suppressor cells.

**Figure 8 cancers-14-04594-f008:**
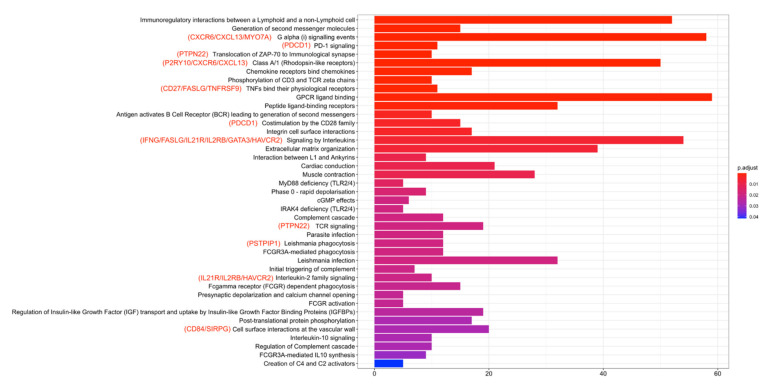
Eighteen CD8+ T_EX_ marker genes were enriched in thirteen pathways, all of which were upregulated in high-low patients. Longer bars indicate that more DEGs were enriched in the pathway. Bar color from blue to red indicates that the DEG-enriched pathways had higher *p*-adjusted values.

**Figure 9 cancers-14-04594-f009:**
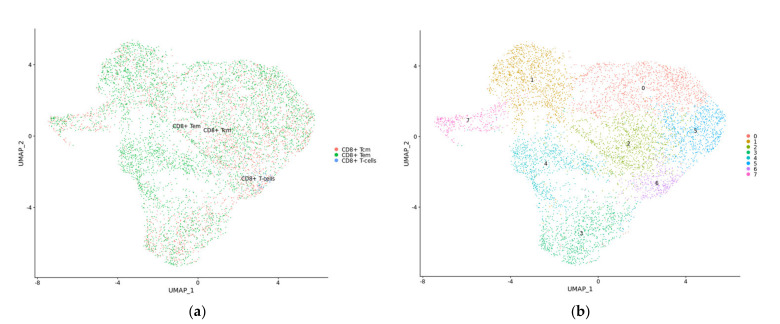
Sub-clusters of CD8T cells from CRC scRNAseq analysis. (**a**) CD8+ T central memory (CM, red), CD8+T effector memory (EM, green), and CD8+ T cell (blue) clusters identified by SingleR subtype identifier. (**b**) Eight distinct CD8+ T cell sub-clusters generated with reclustered CD8+ T_CM_, CD8+ T_EM_, and CD8+ T cells; each colored dot indicates a distinct cluster. The *x*-axis and *y*-axis are two dimensions of the UMAP.

**Figure 10 cancers-14-04594-f010:**
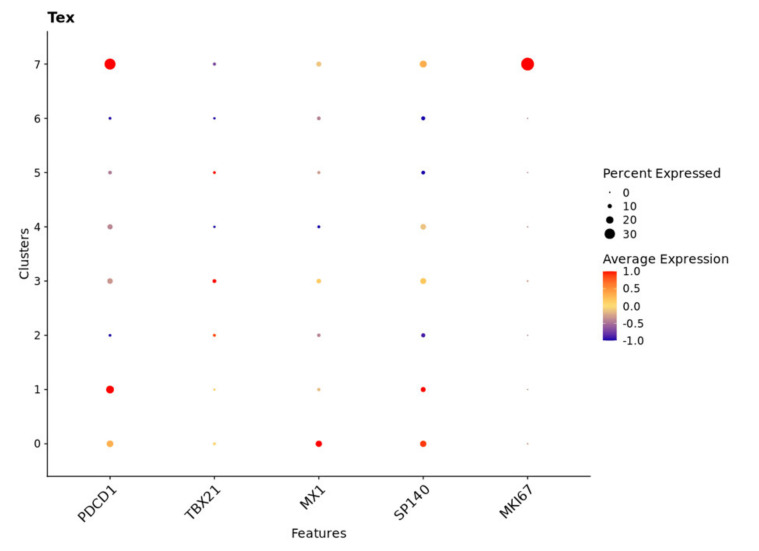
Cluster 1 and cluster 7 were identified CD8+ T cell sub-clusters with CD8+ T_EX_ features. The *x*-axis is the interested gene, with scaled average expression (low to high: −1.0 to 1.0) and a larger dot indicating a higher percentage of expression of interested genes in the total analyzed cells. The *y*-axis is the identified CD8+ T cell sub-clusters.

**Figure 11 cancers-14-04594-f011:**
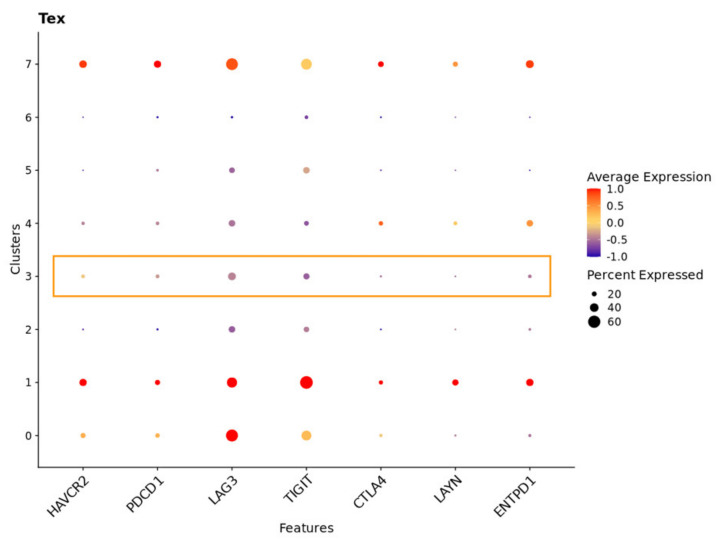
Cluster 3 of CD8+ T cell sub-clusters had the highest TBX21 expression associated with low levels of checkpoint receptors (PDCD1, LAG3, and TIGIT). The *x*-axis is the interested gene, with scaled average expression (low to high: −1.0 to 1.0) and a larger dot indicating a higher percentage of expression of interested genes in the total analyzed cells. The *y*-axis is the identified CD8+ T cell sub-clusters.

**Figure 12 cancers-14-04594-f012:**
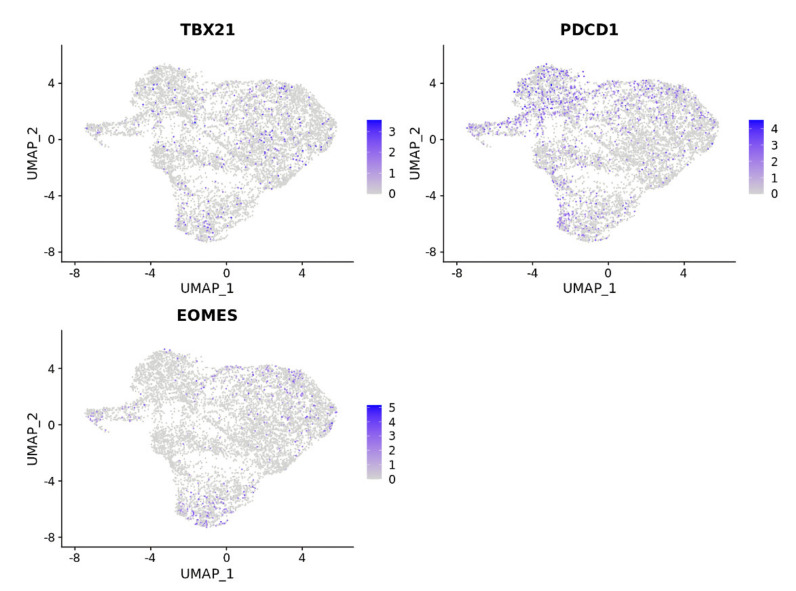
Feature plot to show the distribution of TBX21, PDCD1, and EOMES in each cluster. PDCD1 showed the highest density in CD8+ T_EX_ (clusters 1 and 7) and was associated with CD8+ T_EM_. The *x*-axis and *y*-axis are two dimensions of the UMAP.

**Table 1 cancers-14-04594-t001:** Patient characteristics. CMS, consensus molecular subtype; MSS, microsatellite stable; MSI-L, microsatellite instability-low; MSI-H, microsatellite instability-high.

Characteristic	CMS1 (n = 76)	CMS2 (n = 218)	CMS3 (n = 72)	CMS4 (n = 143)	*p* Value
**Gender, n**					
Male	37	123	38	75	0.6748
Female	39	95	34	68	
**MS, n**					
MSS/MSI-L	14	217	60	137	<0.0001
MSI-H	62	1	12	6	
**Age, Mean ± SD**	71 ± 14	66 ± 12	66 ± 13	65 ± 13	0.0019
**Pathologic stage, n (%)**					
I	12 (16)	46 (21.8)	21 (30.4)	9 (6.6)	<0.0001
II	42 (56)	71 (33.6)	30 (43.5)	49 (36.0)	
III	17 (22.7)	58 (27.5)	15 (21.7)	54 (39.7)	
IV	4 (5.3)	36 (17.1)	3 (4.3)	24 (17.6)	

**Table 2 cancers-14-04594-t002:** Reactome pathway analysis for crosstalk gene with CD8Tex feature from the DEG and DMR analysis of high-low and low-high CMS1 patients.

Pathway Identifier	Pathway Name	Submitted Entities Hit Interactor
**R-HSA-6803204**	TP53 Regulates Transcription of Genes Involved in Cytochrome C Release	TBX21
**R-HSA-6804760**	Regulation of TP53 Activity through Methylation	TBX21
**R-HSA-6804114**	TP53 Regulates Transcription of Genes Involved in G2 Cell Cycle Arrest	TBX21
**R-HSA-6811555**	PI5P Regulates TP53 Acetylation	MX1; SP140; TBX21
**R-HSA-6804758**	Regulation of TP53 Activity through Acetylation	MX1; SP140; TBX21
**R-HSA-6791312**	TP53 Regulates Transcription of Cell Cycle Genes	TBX21
**R-HSA-5633008**	TP53 Regulates Transcription of Cell Death Genes	TBX21
**R-HSA-5633007**	Regulation of TP53 Activity	MX1; SP140; TBX21
**R-HSA-3700989**	Transcriptional Regulation by TP53	MX1; SP140; TBX21

## Data Availability

All the data we applied in this study is publicly available. Synapse repository: https://www.synapse.org/#!Synapse:syn2623706/files/; Cancer Immunome Database: http://tcia.at.

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
