# Peer review of "TBX21 Methylation as a Potential Regulator of Immune Suppression in CMS1 Subtype Colorectal Cancer"

_cancers, 2022, doi:10.3390/cancers14194594_

Round 1

Reviewer 1 Report (Previous Reviewer 1)

Perhaps I'm old-fashioned, but a scientific report in my opinion should contain all the elements of the scientific method. This manuscript contains only analysis of data. The authors then draw correlations and generate a hypothesis that remains untested. There are no new experiments. There is new analysis of existing data from existing experiments. The correlative results make associations that have been previously described in other systems. While these associations are new to CRC, they remain mere associations. The methylation of TBX21 is implied by the authors to regulate T cell exhaustion in CMS1 cancers and impact patient survival, but there have been no mechanistic experiments to show cause/effect. Without mechanistic evidence, it cannot be stated that TBX21 methylation impacts survival. Maybe it does, but there are plenty of other explanations for the correlation. For what it is worth, CRC models using immunocompetent syngeneic mice do exist. See your reference Beltra 2020 Immunity and how they used MC-38 murine adenocarcinoma cells implanted in C57Bl/6 mice (which can be done orthotopically). I believe in order to publish this manuscript, either the language needs to be culled to eliminate any implication of cause/effect, or some type of new experiment should be done to test the hypothesis.

Author Response

Reviewer 2 Report (Previous Reviewer 3)

The authors have nicely revised the original submission.

Author Response

Thank you for your comments.

Reviewer 3 Report (New Reviewer)

I am not the expert in bioinformation. But the data looks good to me. I only have one question that any cell base or animal experiments that could prove the results in the manuscript? 

Author Response

Thank you very much for your comments regarding the data. We do believe that there is further work to be done in order to further work out the mechanism of the relationship between TBX21 methylation, T cell function, and patient outcome in CRC patients. We are currently working to answer these questions in our laboratory but given the timeline for response to review, we believe these studies are outside the scope of this manuscript.

Round 2

Reviewer 1 Report (Previous Reviewer 1)

The text has been revised adequately to address my concerns about implying causation without evidence.

Author Response

Thank you.

This manuscript is a resubmission of an earlier submission. The following is a list of the peer review reports and author responses from that submission.

Round 1

Reviewer 1 Report

In this manuscript, the authors present analysis of publicly available datasets to suggest a role for the TBX21 (T-bet) gene in regulating T cell exhaustion in an inflammatory subtype (CMS1) of colon cancer. Unfortunately, these analyses are all correlative, and there are no new experiments presented. It seems necessary to me that some kind of novel experiments be performed to validate the findings and show (or at the very least suggest) a mechanism for TBX21 function in the context of CMS1 CRC. While the data analysis may be sophisticated, the plots are not presented well. In the available PDF, the figure color schemes, fonts, sizes, spacing, and styles are inconsistent and difficult to follow. There are no figure legends. A reasonable set of work has been published on the role of TBX21 in CD8 T cell exhaustion. The authors should consider their analysis in the context of published work. Here are just a few examples: Yang 2020 Cell (PMID: 33296702), Joshi 2007 Immunity (PMID: 17723218), Sen 2016 Science (PMID: 27789799), and Seo 2021 Exp Mol Med (PMID: 33627794). Most importantly, a very thorough and careful mechanistic dissection of TBX21 and Tex progression has already been described in viral infection and melanoma, see Beltra 2020 Immunity (PMID: 32396847). It is critical for the journal editors and authors of the present study to recognize prior work and consider whether or the presented analyses are novel.

Reviewer 2 Report

Thank you for the opportunity to review this paper. There are some novel findings here. Comments:

  1. The abstract would benefit from some major rewriting. I think it could be overall rewritten for clarity of flow and thought. It would work better if conclusions and conjecture were after the results instead of intermixed. A few additional things:
    1. Line 29, please clarify that this finding is from the TCGA 
    2. Line 29, we found that... should be rewritten for clarity.
    3. Line 35, 'due to...' is out of place here.
    4. In terms of discussing high-high to low-high states in the abstract, the meaning is not likely to be immediately clear to the reader. Please clarify high-high/low-high if to remain in abstract.
  2. Lines 73 - would clarify that we are talking about limitations in the treatment of colorectal cancer. Line 84 - might clarify that this is a pre-clinical study.  Line 87 - also, efforts with hypomethylating agents and HDAC inhibitors have failed to improve effects of immune checkpoint inhibitors
  3. Line 192/Figure 3. It is not clear to me exactly how this comparative analysis was performed. Can the authors please expand.
  4. The discussion could overall be shortened with increased focus for optimal ease and interest of the reader.

General questions,

Did the authors look at MSI-H patients from all sub-sets regardless of CMS subtype? Could it be that the findings here are predominantly due to MSI-H tumors, irrespective of CMS classification?

Reviewer 3 Report

Overall, this is an interesting report identifying TBX21 methylation as a critical factor of outcome in CMS1 colorectal cancer patients.

Criticisms:

  1. 172 rectal patients were included in this study. Rectal cancer patients receive frequently neoadjuvant chemoradiation therapy. Such therapy has a very strong influence on immune infiltrations and gene expression. Please exclude rectal patients who received chemoradiation therapy prior to resection.
  2. The authors make a case for TBX21 methylation and gene expression being prognostic in CMS1. This is interesting. However there are many new treatment option for CMS1 patients (check point inhibitors, BRAF inhibitors/MEK inhibitors). Clinically the more interesting group of patients are CMS4 patients, which have the poorest prognosis, with little treatment optios, and where the tumor is  often immune cell 'rich' but 'cold'.  The authors should investigate whether TBX21 methylation is prognostic in particular in the CMS 4 subtype.
  3. Some of the figures require more legends to explain the data shown. Generally, the figures are of poorer quality.